# Genomic Approaches for Monogenic Kidney Diseases: A Comparative Review of Diagnostic Methods and Precision Medicine Implications

Silvia Giovanella [1], Giulia Ligabue [2], Johanna Chester [2] and Riccardo Magistroni [2,*]

1. Clinical and Experimental Medicine PhD Program, University of Modena and Reggio Emilia, 41124 Modena, Italy; silvia.giovanella@unimore.it
2. Surgical Medical and Dental Department of Morphological Sciences, University of Modena and Reggio Emilia, 41124 Modena, Italy; johanna.chester@unimore.it (J.C.)
* Correspondence: riccardo.magistroni@unimore.it

**Abstract:** Chronic kidney disease is a long-term condition with significant implications for quality of life and health care costs. To uncover the etiology in selected cases suspected of monogenicity, a genomic approach can be employed. There are multiple technologies available, but there is currently no consensus on the most effective diagnostic approach. This review provides a comparison of currently available diagnostic methods in terms of diagnostic yield. However, the heterogeneity of patient cohort inclusion criteria limits direct comparisons. Our review identified three studies which compared a targeted gene panel and whole-exome sequencing for the same patient population. However, the results are inconclusive due to the different sizes and specificity of the targeted panels employed. The contribution of a whole-genome sequencing approach is highly debated. It is noteworthy that a large number of data are generated by these sequencing technologies. This allows for rapid analysis of coding and non-coding regions. However, the interpretation of variants is a significant burden, and the reporting of incidental findings is still challenging. Therefore, the identification of the most efficient technology is pivotal but still controversial. To conclude, an objective comparison of the three methods for the same population could overcome the limits of these studies' heterogeneity and highlight the weaknesses and the strengths of individual approaches.

**Keywords:** genetic testing; CKD; diagnostic yield; review; diagnostic approach

## 1. Introduction

Chronic kidney disease (CKD) is a global public health problem with the adverse outcomes of kidney failure, cardiovascular disease and premature death. Kidney failure (KF), which is the final stage of CKD, affects approximately 9–13% of the world's population, corresponding to 800 million individuals [1–3]. In 2010, approximately 2618 million people worldwide received renal replacement therapy (RRT) [4]. This is predicated to increase to 5.4 million people by 2030 [5]. RRT impacts quality of life and life expectancy [6] and results in substantial health care costs [7].

To address this issue, the implementation of screening and medical investigation strategies for individuals at high risk of developing KF is believed to be crucial [8]. The main reported causes of KF are diabetic nephropathy (19%) [9], glomerulonephritis (17%) and hypertensive nephropathy (16%), but in 20% of RRT cases, the etiology remains undetermined [10].

Recent studies have shown that a genomic approach can detect the underlying cause of CKD in approximately 30% of pediatric and 5–30% of adult cases [11–13]. More than 600 monogenic genes correlated to kidney diseases [14] can be identified. Among the population with early-onset CKD (before 25 years of age), 70% of the diagnoses are congenital anomalies of the kidneys and urinary tracts, cystic diseases, glomerulonephritis and steroid-resistant nephrotic syndrome (SRNS) [15].

A genetic diagnosis has important implications for the diagnostic and therapeutic pathways of a nephropathic patient. More precise prognoses and therapies can be achieved whilst avoiding unnecessary and potentially harmful invasive medical investigations (e.g., renal biopsy) and futile therapies (e.g., immunosuppression). Additionally, genetic diagnosis allows genetic counseling, especially in the family planning stages.

There are several approaches to genomic analysis, each with different diagnostic sensitivities and cost-effectiveness [16]. The most common approaches are based on three main technologies, including targeted gene panels, whole-exome sequencing (WES) and whole-genome sequencing (WGS). Sanger technology remains a validation system for next-generation sequencing (NGS) and for the analysis of small specific genes.

However, there is a lack of consensus in the scientific community on the best diagnostic approach.

Studies reporting the diagnostic yield of genomic diagnostic technologies vary considerably because of methodological inter-study heterogeneity. The target patients range from those with specific clinical presentations, such as SRNS, renal stones, or polycystic kidney disease, through to non-specific CKD clinical presentation and high suspicion of genetic conditions. Further, most studies are retrospective, and very few present direct comparisons of the diagnostic technologies for equivalent target populations. As a result, a conclusive evaluation of the best approach remains elusive.

We aimed to review and evaluate the diagnostic yields of currently employed genomic testing technologies in the field of genetic kidney disease to assist professionals in selecting the best precision medicine approach.

## 2. Genetic Testing: The Current Framework

A review, including original studies and reviews reporting the diagnostic yield of the genomic approach for kidney diseases, was performed. A Pubmed advanced search was conducted, which identified a total of 102 studies published from 2000 to July 2023. The MESH keywords used in the search were kidney disease, diagnostic yield and genetic test. Seven articles were additionally identified and included.

Screening criteria specified the exclusion of studies not reporting the diagnostic yield of the genomic test (30 studies), including <20 patients (20 studies), focused on population screening (3 studies) or renal cancer (8 studies) or reporting on patients already included in previous studies (1 study). Selection based on inclusion and exclusion criteria was performed using the platform Rayyan.

A total of 47 studies were considered and analyzed. Each study's chosen technology, patient cohort characteristics and diagnostic yield were recorded in a dedicated spreadsheet (Table 1). According to the patient cohort age, studies were grouped as pediatric, pediatric/adult or adult only, and as specific or non-specific CKD clinical presentations. A subset of three studies, which included direct comparisons between two different sequencing approaches, was created. Fisher's exact test was used to evaluate the incremental rate of the diagnostic yield for each technology.

The most frequently discussed specific clinical presentations were cystic diseases, focal segmental glomerulosclerosis and SRNS, and tubulopathies. Technologies identified in this review included Sanger sequencing (2 studies), targeted gene panel testing (20 studies), WES (16 studies) and WGS (1 study). A total of 8 retrospective studies included assessments performed using a targeted gene panel and WES but did not provide separate outcomes.

Figure 1 shows the heterogeneity of the diagnostic yield distribution for the different technologies. No formal statistical analysis could be performed, due to the wide differences and variety of the studies collected.

**Table 1.** Summary of included studies, grouped according to specific and non-specific chronic kidney disease (CKD) clinical presentations, age of the included cohort and reported technologies (Sanger; Targeted panel; WES—whole-exome sequencing; Mixed; WGS—whole-genome sequencing). Mixed considers results from targeted panel and WES.

| CKD Clinical Presentation | Technologies | Pediatric | | Pediatric/Adult | | Adult | |
|---|---|---|---|---|---|---|---|
| | | N° Cases | Yield (%) | N° Cases | Yield (%) | N° Cases | Yield (%) |
| Specific | Sanger [17,18] | 35 | 14 | - | - | 2034 | 2 |
| | Targeted panel [19–30] | 31–1554 | 24–78 | 34–859 | 18–81 | 81–236 | 7–22 |
| | WES [31–34] | 24–60 | 42–58 | - | - | 24–193 | 11–36.5 |
| | Mixed [35] | - | - | 45 | 64 | - | - |
| | WGS [36] | | | 144 | 70 | | |
| Non-specific | Sanger | - | - | - | - | - | - |
| | Targeted panel [37–44] | 188–832 | 28–40 | 50–1007 | 21–65 | 135–416 | 12–56 |
| | WES [45–56] | 104–1000 | 32.5–52 | 80–174 | 30–51 | 92–3315 | 9.3–34 |
| | Mixed [57–63] | 158 | 51 | 74–309 | 31–57 | 231 | 42 |

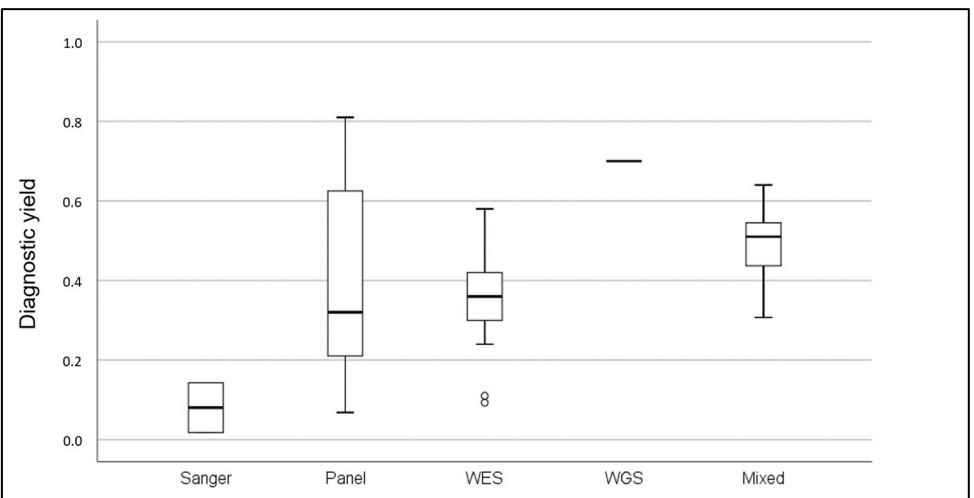

**Figure 1.** Boxplot for the distribution of diagnostic yield among the study included in the review. WES, whole-exome sequencing, WGS, whole-genome sequencing.

### 2.1. Sanger Technology

Sanger technology is a rapid and cost-effective method, allowing for population screening for a single disease [16,64]. It has high coverage and high sensitivity for detecting single-nucleotide polymorphisms. However, this analysis is limited to single DNA fragments of up to 1000 base pairs and is not efficient for large or multi gene analyses. Sanger is useful for the validation of variants identified in NGS, gap-filling in WES technology and analysis of small genes correlated to targeted diseases [16]. It is used for the identification of GLA gene (*Galactosidase Alpha*, 429 amino acid) in suspected Fabry's disease, specifically for the identification of the variant position, thereby determining the best therapeutical approach. In Fabry disease, the diagnostic yield is reported to be approximately 14% in pediatric and 2% in adults [17,18] (Table 1).

### 2.2. Targeted Panel Technology

Targeted panel can assess multiple genes in parallel, saving time and reducing costs. Targeted panels focus on a selected set of genes with known or suspected associations with the disease under study. Employing panel technology is an effective approach to establishing a genetic diagnosis, in case of a clearly defined phenotype, to define the molecular diagnosis between phenocopies [52]. In the case of a defined phenotype and limited

genes to test, the approach is more simple than WES or WGS approaches. Furthermore, employing a specific genes panel helps prevent the discovery of incidental findings, as there is currently no consensus on how to report them.

On the other hand, the targeted panel cannot be easily extended to include new genes. This means that the panel's content might quickly become outdated over time as new genetic associations are identified. Updating panels to include new genes associated with the disease or phenotype, requires a new design and validation of the modified panel [65]. Furthermore, in consideration of the multitude of clinical presentations, a significant number of different panels should be maintained by the analytical laboratory, thereby increasing the complexity and cost of this solution.

Currently, targeted panels vary in the number of genes included in the analysis, with the largest panels able to analyze approximately 2000 genes. The clinical impact largely depends on the genes included and therefore, whether a differential diagnosis can be made.

Table 1 reports diagnostic yields for targeted panels which are disease-specific, ranging from 6.8% for patients with renal stones [28] to 81% for patients with a clinical diagnosis of Autosomal Dominant Polycystic Kidney Disease (ADPKD) [26]. In the case of non-specific CKD clinical presentations, diagnostic yield from panels range from 28 to 40% for pediatric cases [37,38], 21 to 65% for pediatric/adult cases [39–41] and 12 to 56% for adult cases.

Most disease specific studies have used small panels (genes < 100), reporting an average yields of 36%. Only two targeted studies used a larger panel [20,22], both on cystic populations, with an average yield of 75%.

The difference in yield, however, is largely related to the clinical presentation investigated: three studies on cystic patients observed an overlapping yield (70–80%) independent of whether the study was conducted on a small or a broad-spectrum panel. In contrast, there are disorders, such as hypophosphatemia, in which a panel of 13 genes leads to a yield of 63% [25] and renal stones disorders in which a panel of 45 has a yield of 7% [28].

Small panels have not been employed in case of non-specific clinical presentations.

### 2.3. Whole-Exome Sequencing

The WES approach captures the majority of genomic coding regions. It is estimated that up to 85% of all pathogenic mutations fall within this region. WES is mainly applied in cases of unclear clinical suspicion. This technology enables the analysis of targeted genes, whilst maintaining information for future "virtual panel" analyses as new genomic components are identified. The employment of a WES based virtual panel reduces the computational burden and restricts the occurrence of incidental findings, while preserving data for reanalysis as additional genetic discoveries emerge [56]. WES exhibits low coverage compared to other methods and often requires Sanger sequencing to increase the depth of analysis [16]. WES does not achieve a complete coverage of the exome in some areas, such as GC-rich regions and homology sequences [66]. The GC content can reduce the efficiency of nucleotide hybridization, causing lower sequencing coverage [67,68] and requiring modified Polymerase Chain Reaction (PCR) conditions and the design of exon-specific primers. However, the employment of long-read sequencing technology in the WES approach (third-generation sequencing), may overcome the low coverage challenge and enhance sensitivity, especially in high-homology regions [69].

WES technology remains a cost-effective option compared to WGS [70], with simplified data analysis and storage. Jayasinghe et al. reported that, compared to kidney biopsy, WES is a lower risk and more cost-effective approach for children with a glomerular disease [31].

Table 1 highlights the heterogeneous diagnostic yield. For pediatric populations with specific CKD clinical presentation, yields range from 42 to 58% [31,32], and in adults the yields range from 11 to 36.5% [31,33,34].

The corresponding diagnostic yield for non-specific CKD clinical presentations range from 32 to 52% for pediatric cases [45–48], 30 to 51% for pediatric/adult [49–52] and 9.3 to 34% for adult cases [53,54,63].

Several retrospective studies did not provide separate outcomes from targeted panel or WES. In these cases, the yield for non-specific clinical presentation patients is 51% in pediatric [57], 31–57% in pediatric/adult [55,58–60] and 42% in adult cohorts [56,61].

Studies Including Direct Comparisons of 2 Technologies

We identified three studies [54,71,72] that directly compared the diagnostic yield of two different technologies for the same population (Table 2).

One study [71] reported no statistical differences between an enlarged panel (2703 genes, ClearSeq Inherited Disease Panel-Agilent, Santa Clara, CA, USA) and WES. The other two studies [54,72] show a significant incremental rate (5% and 10%) of WES compared to targeted panel.

The main difference between the 3 studies is expressed by the type of targeted panels used. Rao et al. [71] conducted analyses on a large panel of 2703 genes, whereas Wilson et al. [72] and Groopman et al. [54] employed smaller, selected phenotype-specific targeted panels. Wilson et al. and Groopman et al. both reported significant differences in diagnostic yield between targeted panels and WES. However, diagnostic yield is directly associated with the correct selection of a targeted panel based on clinical presentation. Therefore, incorrect initial diagnostic suspicion can lead to the selection of an incorrect targeted panel, which a broader WES analysis can correct. In the Wilson et al. [72] study, the diagnoses identified through WES analysis only, are on genes included in several "expanded panels". Notably, all but 1 are found in the expanded targeted panel employed by Rao et al. [71].

A clinical misdiagnosis or genetic heterogeneity of some conditions can consequently result in the utilization of inappropriate disease-specific panels and misdiagnosis. With the selection of broad panels or genome-wide approaches, this risk is minimized. As reported in several studies, following genetic testing, between 10 and 45% of cases [22,42,54,55,73], undergo a reclassification of the original clinical diagnosis. A WGS approach maximizes diagnostic potential, although this strategy does not yet have immediate applicability in routine diagnostics.

**Table 2.** Comparison of technologies' yield in the same population setting. WES, whole-exome sequencing.

| Technology | N° Cases | Diagnostic Yield (%) | Incremental (%) | References |
|---|---|---|---|---|
| Panel (2703 genes) | 482 | 42.6 | No statistical differences ($p > 0.05$) | [71] |
| Singleton-WES | 196 | 36.2 | | |
| Trio-WES | 317 | 44.8 | | |
| Panels (virtual analysis) | 3315 | 4.1 | 5 ($p < 0.01$) | [54] |
| WES | 3315 | 9.3 | | |
| Panels (disease specific) | 324 | 20.0 | 10 ($p < 0.01$) | [72] |
| WES | 324 | 30.0 | | |

*2.4. Whole-Genome Sequencing*

The most advanced technological approach is WGS, which allows for the investigation of the entire genome, including regulatory regions and non-coding variants via short- or long-read technology.

It has good coverage, an improved mappability compared to WES and no GC-bias, reducing rates of false-negative variant calls [67]. Table 3 summarizes the list of advantages and disadvantages of applying WES or WGS.

The WGS approach overcomes the pseudogene sequence similarity and duplicate regions [36]. These characteristics are especially important for some diagnoses, such as ADPKD [74]. The PKD1 (*Polycystin-1*) gene shows six pseudogenes with a sequence similarity of 97%. Mallawaarachchi et al. [36] report a diagnostic yield of WGS in ADPKD patients of 80%. A similar yield was reported by Bullich et al. [20] in a cohort of clinically

diagnosed ADPKD patients. In this study, patients were screened with a panel of 140 genes. The only limitation of this approach is the redesigning and revalidating of the setting, each time a new gene is discovered.

The WGS short-read technology approach, still largely adopted in next-generation systems, is struggling to detect complex structural variants such as large inversions, deletions, or translocations. The technology advancement of long-read technology could overcome this limitation [75].

Despite its potential, WGS is still not widely used in clinical practice due to its higher costs and less developed analytical tools compared to more established sequencing methods, such as WES. However, with ongoing advancements and cost reductions, WGS may become more commonly employed in the future, especially for cases where its unique benefits are crucial, such as diagnosing complex genetic conditions like PKD.

Nevertheless, the requirement of dedicated infrastructures, trained specialists, functional studies for uncertain significance variants (VUS) interpretation is a challenge for clinical centers and for large-scale implementation of these approaches [76]. An extensive application of WES would lead to the multiplication of the VUS to be documented. The problem is exacerbated by WGS where huge portions of the non-coding genome are effectively uninterpretable. From a research perspective, the availability of large amounts of data has led to the establishment of large-scale WGS projects, such as the European program Beyond 1 Million Genomes [77]. However, in a routine clinical context, a panel approach is a more effective and manageable implementation. These data will contribute to the advanced identification of rare pathogenic variants in the coding and non-coding domain.

Currently, studies reporting the diagnostic performance of WGS in the field of kidney diseases are few and none met our study's inclusion criteria. Various non-nephrological studies have reported an increase in the effectiveness of WGS, but results are conflicting. Bertoli-Avella [78] et al. in a study including 1007 patients with neurological diseases, demonstrated that 30% of the unsolved WES cases could benefit from WGS. However, the interpretation of these data is not unique. Biskup et al. [79], in a responding letter to the Bertoli-Avella et al. study, reported that the genomic variants identified by WGS can also be identified using a well-established WES technology, suggesting that only 1.4%, and not 30%, of the genomic variants can be identified by the WGS approach.

**Table 3.** Advantages and disadvantages of whole-exome sequencing and whole-genome sequencing.

| | Advantages | Disadvantages |
|---|---|---|
| Whole-Exome Sequencing | Detection of the coding-region variants. | Low coverage, mainly in GC rich and homology sequences. |
| | Applicability in unclear CKD case. | Need for gap filling with Sanger. |
| | Cost effective options compared to WGS. | Burden of incidental findings. |
| | Possibility to create virtual panel of analysis to reduce the burden of VUS variants detected. | |
| Whole-Genome Sequencing | Detection of the coding-region and non-coding variants. | Higher costs compared to WES. |
| | Detection of structural variants. | Need for dedicated infrastructure and trained specialists. |
| | Applicability in unclear CKD case. | Relevant burden for VUS variants and incidental findings. |
| | Ability to detect CNVs and variants in high-homology regions (PKD1): one strategy for all the detections. | The superiority in diagnostic yield is debated (from 2% to 30%). |

In other studies, the incremental rate between WES and WGS varies from 2 to 9% [69,80]. Evans et al. [69] reported an additional 9% yield of WGS with 13 new diagnoses in cases previously unresolved by WES analyses. These additional detected diagnoses were related to unknown gene-disease associations, insufficient sequencing coverage and copy number variations (CNVs).

WGS offers advantages, including high coverage and the ability to detect CNVs. In the context of genetic kidney disease, where genes such as PKD1 and MUC1 (*Mucin 1*) pose challenges due to high-homology regions and variable-number tandem repeats (VNTR) [81], WGS allows for comprehensive screening of all genes and associated variants. However, the routine implementation is hindered by challenges such as the need for expertise, infrastructure, variant interpretation and cost, making it a significant burden.

### 2.5. Ethical Implications of Genetic Testing

The increased amount of genomic data to be analyzed brings risks [14]. Potential incidental findings, now referred to as "secondary findings" by ACMG [82], require a reanalysis of clinical investigations and close clinical follow up, initiating a new "diagnostic odyssey" for the patient. The lack of consensus on reporting these findings limits the widespread use of WES and WGS approaches.

In 2013, the ACMG released a list of 59 genes (updated to 73 [83]) that should be disclosed as secondary findings to mitigate the risks associated with specific highly penetrant genetic disorders [82].

The increasing amount of genetics data raises ethical and privacy issues [84]. Returning diagnostic genetic data and secondary findings may cause patient anxiety, interpretation challenges for the clinician and may carry ethical responsibility.

In the United States, the patient often has the option to request the exclusion of secondary findings. The European Society of Human Genetics recommends reporting of secondary findings, whilst emphasizing the importance of patient autonomy and the inclusion of options for expressing individual preferences in the informed consent. Other states have kept a more conservative approach. However, it is crucial that the patient is aware that secondary results may be returned.

Patients may also face discrimination in employment and insurances [65]. Europe and the U.S. have taken measures to address this issue, such as the publication of the Recommendation CM/Rec(2016)8 [85] and Genetic Information Nondiscrimination Act (GINA).

It is essential for researchers, geneticists, clinicians, governments and society to address these ethical concerns to ensure that genomic sequencing benefits individuals and society while respecting individual rights and privacy.

### 3. Discussion

This review highlights a heterogeneity of diagnostic performance within the single technologies, rendering any comparison of technologies difficult to make.

A targeted gene panel is a convenient and largely available technology in the clinical setting. It is mainly used to evaluate patients who are clinically well characterized and can assess from 10 to over 2000 genes. The diagnostic yield of the targeted panel can reach 80% in ADPKD patients [26].

However, targeted panels can become quickly outdated as new genes linked to specific clinical presentation are continuously identified, especially in the case of heterogeneous or not well clinically defined phenotypes. Initial clinical indication is not always confirmed by genetic testing, and diagnoses are changed or reclassified after genetic testing in up to 45% of cases [22]. These limitations have encouraged numerous centers around the world to adopt the WES approach, filtering for genes of interest. This makes it possible to update the list of genes to be filtered or, when appropriate, to analyze in the future.

However, WES technology has a low coverage in GC-rich and homology regions, requiring an adjustment of PCR conditions. WGS overcomes this limitation, identifying structural and non-coding variants. The employment of a single method to screen patients

for single or copy number variations and structural variants on genes, such as MUC1 or HNF1b [81,86], enables the identification of the main causes of disease. However, the high costs of WGS technology and the difficulty of data analysis make it less likely to be adopted outside of research settings [87].

The adoption of genomic methods has significantly increased the number of variants to be evaluated, raising the issue of secondary findings [82]. Currently, there is no global consensus on the management of secondary findings.

A feasible, current solution involves the inclusion of panels or virtual panels focused exclusively on known diagnosis-related genes. In the case of negative results, research can then be extended [35,56]. The integration of multiple analytical methods seems to show a better average yield (Figure 1). In the scenario of increasing prevalence and demand for genomic sequencing, the identification of the most efficient and cost-effective technology is pivotal. However, the heterogeneity of the studies and the absence of direct comparison between available technologies in similar study settings, limits comparative evaluations.

Table 1 reports the range of diagnostic yields reported among studies included in this review. There are no correlations between the yield and the technology used or patient age. However, no formal statistical analysis could be performed, due to the important heterogeneity of the studies meeting our inclusion criteria, type of population tested and the number of cases. Among studies of direct comparison of two technologies, WES was superior to targeted panels. However, these data may be confounded by the use of inappropriately selected disease-specific panels. Had a much broader panel, including 2700 genes been used, misdiagnoses due to heterogeneous phenotypes, phenocopies or misclassification of pathology, may have been avoided [71].

Despite the high heterogeneity and inconclusive, single study comparison of two technologies, we hypothesize that the highest diagnostic yield can be found in a well selected targeted panel strategy adopted for patients with specific clinical presentation, such as ADPKD [20,26]. Conversely, in patients with non-ADPKD clinical presentation, such as renal stones and Bartter disease, the diagnostic yield is lower, ranging from 6.8% and 17.7% [27,28].

In the case of WES technology, the results are extremely variable. The diagnostic yield reaches 42% in pediatric/adult setting in a study of 102 patients affected by CKD [50], but it is only 9.3% in a study that involved 3315 adults [54].

It is possible that new technologies, such as single-molecule sequencing, which relies on longer-read sequencing (>10 kb vs. less than 1000 bp in short-read), may overcome the limitations associated with gap filling in WES and faster determination of CNVs. These emerging technologies hold significant promise. They not only offer improved accuracy in identifying structural variants but also extend the ability to analyze regions that were previously inaccessible using short reads. It will be interesting to assess the applicability of these technologies and whether their diagnostic performance can exceed that of WGS and WES [88].

## 4. Conclusions and Future Prospections

The lack of consensus in the scientific community about the best diagnostic algorithm for genetic kidney disease is mainly due to the heterogeneity of patient selection, study settings and technological methodology within reported studies, therefore restricting any viable comparisons.

Two studies directly employed two technologies for the same patient population in the same design setting, and they reported the superiority of WES compared to targeted gene panels in terms of diagnostic yield. The advantage of future analysis of data and, therefore, reclassification of diagnosis, is a major advantage of exome and genome-wide approaches that cannot be performed with targeted panels.

However, the huge amount of data produced by WES and WGS can lead both to the discovery of new disease-associated pathways and new disease genes, but also to genetic misdiagnosis and unnecessary referrals.

This review provides the current level of evidence and narrative comparison of diagnostic yields of the technologies currently employed in genomic diagnoses of kidney diseases. Future research should be conducted in homogeneous patient populations, considering well clinically characterized patients. It would be helpful to analyze the same large population with the three approaches; a targeted panel containing the main genes related to genetic kidney disease, WES and WGS to enable an objective comparison of the methodologies. However, the substantial costs associated with such a study design continue to pose a prohibitive barrier to realization.

Currently, technologies that enhance performance and increase the potential to detect genetic variants increase the chances of correctly referring patients toward new therapies and counseling services. However, technologies that investigate the human genome, will increasingly face ethical and potential discrimination issues.

**Author Contributions:** Conception and drafting the article: S.G. and R.M. Analysis and interpretation of data: S.G. and R.M. Revising the article: R.M., J.C. and G.L. All authors have read and agreed to the published version of the manuscript.

**Funding:** This research received no external funding.

**Informed Consent Statement:** Not applicable.

**Data Availability Statement:** Not applicable.

**Conflicts of Interest:** The authors declare no conflict of interest.

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
