# Peer review of "Genomic Approaches for Monogenic Kidney Diseases: A Comparative Review of Diagnostic Methods and Precision Medicine Implications"

_applsci, doi:10.3390/app132312733_

Round 1

Reviewer 1 Report

Comments and Suggestions for Authors

Giovanella et al. have reviewed the genetic testing landscape for monogenic kidney diseases, and attempted to compare the various options.  This is a very important topic, as chronic kidney disease places a major burden on the healthcare system, and the nephrology field is only now beginning to embrace the importance of monogenic diseases as a cause of CKD.  There are a number of hurdles to wide spread implementation of genetic testing, one of which is good data comparing the genetic yield rate of various genetic testing methods that would allow physicians to make an educated choice.  And as the authors point out, the heterogeneity of CKD cohorts make this difficult.

Based on the inclusion criteria for articles, there are a number of articles that I am aware of that I believe should have been included in the analysis.  I would request that these either be included, or else please point out which exclusion criteria they satisfy:

Connaughton DM, Kennedy C, Shril S, et al. Monogenic causes of chronic kidney disease in adults. Kidney Int. Apr 2019;95(4):914-928.

Murray SL, Dorman A, Benson KA, et al. Utility of Genomic Testing after Renal Biopsy. Am J Nephrol. 2020;51(1):43-53.

Mansilla MA, Sompallae RR, Nishimura CJ, et al. Targeted broad-based genetic testing by next-generation sequencing informs diagnosis and facilitates management in patients with kidney diseases. Nephrol Dial Transplant. Jan 25 2021;36(2):295-305.

Schrezenmeier E, Kremerskothen E, Halleck F, et al. The underestimated burden of monogenic kidney disease in adults waitlisted for kidney transplantation. Genet Med. 2021 Jul;23(7):1219-1224.

Ottlewski I, Munch J, Wagner T, et al. Value of renal gene panel diagnostics in adults waiting for kidney transplantation due to undetermined end-stage renal disease. Kidney Int. Jul 2019;96(1):222-230.

Mallett AJ, McCarthy HJ, Ho G, et al. Massively parallel sequencing and targeted exomes in familial kidney disease can diagnose underlying genetic disorders. Kidney Int. Dec 2017;92(6):1493-1506.

Elhassan EA, Murray SL, Connaughton DM, et al. The utility of a genetic kidney disease clinic employing a broad range of genomic testing platforms: experience of the Irish Kidney Gene Project.  J. Nephrol. 2022 Jul;35(6):1655-1665.

Maallawaarachchi AC, Lundie B, Hort Y, et al. Genomic diagnostics in polycystic kidney disease: an assessment of real-world use of whole-genome sequencing. Eur. J. Hum. Genet. 2021 May;29(5):760-770.

Of these papers, Mallet et al., Elhassan et al., and Maallawaarachchi et al., perform some comparisons between methods, and the latter one looks at WGS

While it falls outside of the time window in the inclusion criteria, here is another relevant, very recently published article:

https://journals.lww.com/jasn/abstract/9900/the_clinical_utility_of_genetic_testing_in_the.195.aspx

Targeted panel technology can be applied in different ways, from small phenotype specific panels to broad panels that attempt to encompass all CKD-related genes.  A challenge with the smaller phenotype-specfic targeted panels is the fact that phenotypes of CKD-related genes can be very heterogeneous, and clinical diagnoses are often inaccurate.  There is a reasonable possibility that smaller panels will miss causative genes in CKD patients who have what appear to be clearly defined disease. Broad panels (along with WES/WGS approaches) help to address this issue. For example, in Groopman et al. more than 10% of patients with a genetic finding reclassified the initial diagnosis. This difference should be discussed. 

Following this discussion, Figure 1 shows that panel yield rates have a very large range.  This is likely due to the fact that smaller panels and larger panels have been lumped together.  While there is no clear distinction between small and large panels, I would recommend that the authors  separately analyze papers reporting on smaller and larger panels.

One reason that WES and especially WGS is slower to be adopted is practical:  there is a higher burden for variant curation and genetic counseling.  There is much less information about the phenotypic associations with non-exomic regions, leading to a much higher likelihood of variants of unknown significance.  While this may be easier to accomplish by academic centers that have the relevant expertise, it is much more difficult to implement at scale.  The authors touch upon this in the discussion, noting the more difficult of data analysis. A few more sentences would be welcome.  

This review covers an important topic, and should be published once the authors revise the manuscript.

Reviewer 2 Report

Comments and Suggestions for Authors

The authors have presented an interesting review of the diagnostic methodologies utilized for establishing the genetic basis of renal diseases. I must commend them on a good job.

I would like to highlight a few points that I think the authors should consider:

1. Targeted genome sequencing or genetic panels are useful for patients who have been robustly phenotyped and fulfil specific genetic/diagnostic criteria. However, this method hinders new gene discovery and may lead to misinterpretation of variant pathogenicity. 

2. WGS should be the gold standard. I agree with the authors about the potential mishandling of bulk datasets, but the coverage of whole genome sequencing, allows for copy number variation analysis and variant detection in non-coding, deep intronic regions. 

I would encourage the authors  to comment on the above in their discussion and highlight the possibility that intronic and splice site variant analysis of whole genome data may be a game changer in terms of diagnostic yields. 

Comments on the Quality of English Language

Minor edit: LINE 243 - Heterogenicity should be changed to heterogeneity

Reviewer 3 Report

Comments and Suggestions for Authors

Dear Auther

Here are my comments on the provided manuscript:

General Comments:

  1. Title: The title is clear and appropriately reflects the content of the paper.
  2. Abstract: The abstract gives a brief overview of the topic and sets the scene for the review. It could benefit from a brief mention of the main findings or conclusions to better inform the reader about the outcomes of the review.

Specific Comments:

  1. Introduction:
    • Lines 25-31: The global prevalence and impact of CKD are well established. Consider adding references for the statistics provided.
    • Line 36: The term "etiology is missing" is vague. Consider rephrasing to "the etiology remains undetermined" or "the cause is unknown."
    • Please add the following citation Al-Awaida WJ, Hameed WS, Al Hassany HJ, Al-Bawareed O, Hadi NR. Evaluation of the genetic association and expressions of Notch-2/Jagged-1 in patients with type 2 diabetes mellitus. Medical Archives. 2021 Apr;75(2):101.

  2. Genetic Testing: The Current Framework:
    • Lines 68-74: The methodology for the literature review should be detailed more clearly. State the keywords used in the Pubmed search and the criteria for inclusion/exclusion.
    • Lines 85-87: If no statistical analysis could be performed due to heterogeneity, it would be helpful to visually represent the data, perhaps in a figure or graph.
  3. Sanger Technology:
    • Lines 98-103: The information provided about Sanger sequencing is good, but adding references would strengthen the claims.
  4. Targeted Panel Technology:
    • Lines 117-120: It's mentioned that the content of targeted panels might become outdated. Consider discussing the flexibility or lack thereof in updating these panels with emerging genetic information.
    • Line 124: The term "varying" may be better replaced with "ranging from...to..." for clarity.
  5. Whole Exome Sequencing:
    • Lines 137-141: The limitations of WES are well outlined. Consider discussing potential solutions or improvements in technology that might address these limitations.
  6. Whole Genome Sequencing:
    • Lines 189-191: The advantages of WGS are clearly stated. However, more discussion on its limitations, especially in the context of CKD, would balance this section.
    • Lines 207-211: The debate between WES and WGS is a significant point. It might be helpful to have a separate subsection discussing the comparative advantages and disadvantages of both.
  7. Conclusion:
    • The manuscript seems to end abruptly after discussing WGS. A conclusion section summarizing the main findings, discussing their implications, and suggesting future directions would be beneficial.
  8. References:
    • Ensure that all references are correctly formatted and consistently styled. Additionally, ensure all claims and statistics are appropriately cited.
  9. General Suggestions:
    • Consider adding a section discussing the ethical implications of genetic testing, especially concerning incidental findings.
    • Given the rapidly evolving nature of genomic technologies, a discussion or perspective on emerging technologies and their potential impact on the field would be beneficial.

Summary:

  1. Lines 218-220: The initial remark about the heterogeneity of diagnostic performance is clear and sets the stage for the ensuing discussion.
  2. Lines 222-224: It is good to note the constant evolution of the field and the challenges that arise from it. However, the term "obsolete" might sound strong. Consider using "outdated" or "less relevant" instead.
  3. Lines 228-232: The pros and cons of WES and WGS are well-articulated, but it would be beneficial to emphasize the clinical implications of each approach more.
  4. Lines 234-237: The issue of incidental findings is crucial. This section could be expanded upon, discussing ethical implications and how different institutions or countries handle such findings.
  5. Lines 238-240: A critical point is made about the lack of direct comparisons between technologies. This has been a recurring theme throughout the paper and might be a strong recommendation for future research.
  6. Lines 246-248: The discussion of the studies comparing two technologies is insightful. However, referencing the specific technologies in this section would clarify the discussion.
  7. Lines 249-257: The proposed hypothesis about the targeted panel strategy is well-founded and adds to the paper's depth.

Conclusion and Future Prospections:

  1. Lines 259-261: The conclusion starts strong by reiterating the challenges in the field.
  2. Lines 263-266: The advantage of future reanalysis with WES and WGS is a crucial point and is well-presented.
  3. Lines 268-269: The double-edged sword of WES and WGS—potential discoveries vs. misdiagnoses—is well-articulated.
  4. Lines 271-274: The recommendation for future research is clear and actionable.
  5. Lines 275-278: Addressing the financial barriers is important, as it provides context about why certain studies haven't been conducted yet.
  6. Author Contributions: It's good to see a clear division of labor among the authors. This section is well-structured.
  7. Funding through Conflicts of Interest: These sections are standard in scientific papers and seem to be appropriately addressed.

General Comments:

  1. Citations: Ensure that all the citations (e.g., (24), (48), etc.) correspond correctly to the references section.
  2. Language and Clarity: The manuscript is generally well-written. Some sentences could be rephrased for clarity, and there are a few instances where more concise wording could be used.
  3. Recommendations: The manuscript would benefit from a more detailed discussion on the ethical implications of genetic testing, especially with WES and WGS. Additionally, given the rapid technological advancements, a brief mention of potential future technologies or methodologies might be insightful.

Round 2

Reviewer 1 Report

Comments and Suggestions for Authors

The authors have address the concerns raised in my comments.

Comments on the Quality of English Language

A couple of minor typos, but the English is good.

An example that jumped out is in the sentence at the end of section 2.3:  "An issue that does not occur employing broad panels or genome wide approaches". This is a clause, and should be joined to the previous sentence.